# S2TD-Face: Reconstruct a Detailed 3D Face with Controllable Texture from a Single Sketch

## ABSTRACT

3D textured face reconstruction from sketches applicable in many scenarios such as animation, 3D avatars, artistic design, missing people search, *etc.*, is a highly promising but underdeveloped research topic. On the one hand, the stylistic diversity of sketches leads to existing sketch-to-3D-face methods only being able to handle pose-limited and realistically shaded sketches. On the other hand, texture plays a vital role in representing facial appearance, yet sketches lack this information, necessitating additional texture control in the reconstruction process. This paper proposes a novel method for reconstructing controllable textured and detailed 3D faces from sketches, named S2TD-Face. S2TD-Face introduces a two-stage geometry reconstruction framework that directly reconstructs detailed geometry from the input sketch. To keep geometry consistent with the delicate strokes of the sketch, we propose a novel sketch-to-geometry loss that ensures the reconstruction accurately fits the input features like dimples and wrinkles. Our training strategies do not rely on hard-to-obtain 3D face scanning data or labor-intensive hand-drawn sketches. Furthermore, S2TD-Face introduces a texture control module utilizing text prompts to select the most suitable textures from a library and seamlessly integrate them into the geometry, resulting in a 3D detailed face with controllable texture. S2TD-Face surpasses existing state-of-the-art methods in extensive quantitative and qualitative experiments. The code will be publicly available.

## CCS CONCEPTS

• **Computing methodologies** → **Reconstruction**.

## KEYWORDS

3D Face Reconstruction, Face Sketch, Rendering

## 1 INTRODUCTION

Reconstructing 3D textured faces from sketches has been a valuable research topic, finding applications in custom-made 3D avatars, artistic design, criminal investigation, *etc.* However, existing sketch-to-3D-face methods [28, 61] suffer from the following issues. On the one hand, the diverse styles of face sketches, ranging from realistic representations with detailed shading to cartoon-like drawings with simplified lines [60], pose challenges for existing methods that typically focus on sketches with frontal poses [28] or rely

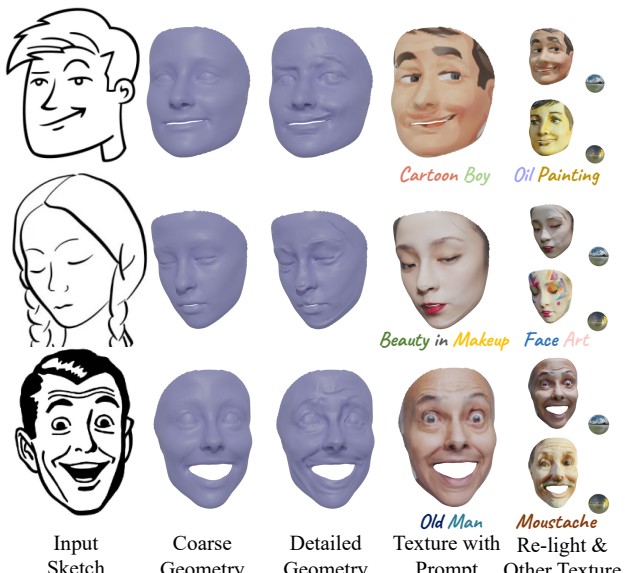

| Input Sketch | Coarse Geometry | Detailed Geometry | Texture with Prompt | Re-light & Other Texture |

**Figure 1: S2TD-Face can reconstruct high-fidelity geometry from face sketches. The texture control module seamlessly applies suitable textures onto the geometry based on prompts. The results can be re-lighted for various application scenes.**

heavily on realistically shaded sketches as input [61]. On the other hand, texture plays a crucial role in accurately portraying facial appearance, highlighting the necessity for texture control within the sketch-to-3D-face process, while existing methods lack this capability. Furthermore, the absence of matching data between sketches and 3D faces makes it hard to train the framework. This paper proposes a method to reconstruct topology-consistent 3D faces with fine-grained geometry that precisely matches the input sketch and allows users to control the texture of reconstruction through text prompts, named **S2TD-Face (S**ketch to controllable **T**extured and **D**etailed **T**hree-**D**imensional **Face**). We introduce S2TD-Face in three parts: geometry reconstruction, training strategies, and texture control module.

One straightforward approach to reconstructing 3D faces from sketches might involve first translating 2D sketches to 2D face images [13, 40, 52], followed by utilizing existing 3D face reconstruction methods [17, 19, 26, 31, 59, 66, 67] to obtain the 3D faces. However, this approach suffers from the following shortcoming. It heavily relies on the cooperation of both the sketch-to-image and image-to-3D-face stages, where inherently sparse but important geometric information like dimples or wrinkles in sketches is often lost during the transformation process, as the two transformation steps are independent, leaving sketches unable to directly constrain the final 3D geometry. In contrast, S2TD-Face uses a direct and efficient geometry reconstruction framework. It firstly predicts the coefficients of 3DMMs [4, 5] from input sketches to reconstruct

coarse geometry and then utilizes coarse geometry and sketches in UV space to generate displacement maps for detailed geometry.

To ensure the framework reconstructs 3D detailed faces that accurately reflect the delicate features of the input, we introduce a novel sketch-to-geometry loss function to supervise both coarse and detailed geometry. This function combines differentiable rendering techniques [34, 51] to extract sketches of different styles from both geometry stages and compare them with ground truth sketches, guiding geometry deformation, as shown in Fig. 4. To ensure the robustness of the reconstruction framework across different sketch styles, we generate 5 different types of sketches for each face image by using traditional filtering operators [6] and deep learning methods [56, 57], as shown in Fig. 2 (a)-(e), with each sample randomly selecting a sketch type as input during training. The framework is trained by 2D signals like landmarks, segmentation, and perception features, as shown in Fig. 2 (f)-(h), without relying on the 3D face scanning data. Based on the widely-used REALY benchmark [9], we tailor it to better suit sketch-to-3D-face tasks for geometry evaluation by transforming the test samples into different styles of sketches, conducting fair evaluation on state-of-the-art methods [17, 19, 20, 26, 28, 37, 54]. Extensive experiments indicate that our method significantly outperforms existing methods.

S2TD-Face controls the texture of reconstructed 3D faces based on a text-image module, offering the following capabilities: it can search for suitable texture from a face library based on the text prompt, transform the selected texture information to UV space, and seamlessly apply the UV-texture to the reconstructed geometry. As shown in the first row of Fig. 1, when the user provides a text prompt describing the desired texture, S2TD-Face can reconstruct 3D textured faces in styles such as 'Cartoon Boy' or 'Oil Painting'.

In summary, the main contributions of S2TD-Face are as follows:

- An effective framework for reconstructing 3D detailed high-fidelity faces from sketches with a novel sketch-to-geometry loss, which accurately captures the local strokes of the input.
- A novel texture control module for controlling the texture of the reconstructions, resulting in textured 3D faces with various styles ranging from cartoons to realistic appearances.
- Extensive experiments show that our method achieves excellent performance and outperforms the existing methods.

## 2 RELATED WORK

**3D Face Reconstruction.** Reconstructing 3D faces from 2D images has achieved widespread success. Methods such as [17, 19, 26, 37, 59, 66, 67] can generate realistic 3D faces from facial images captured in various poses, environments, and expressions. These methods typically utilize 2D landmarks, segmentation, texture information, *etc.* to guide the deformation of 3DMMs [4, 5], and further leverage differentiable rendering techniques [14, 21, 34, 41, 51, 55] for fine-grained reconstruction [19, 37]. These validated 2D-to-3D supervision approaches are applicable for training sketch-to-3D-face framework. Some methods [2, 18, 24, 35, 36] focus on texture reconstruction. They typically decompose textures into components such as diffuse, specular, ambient occlusion, normal, and translucency, applicable in re-rendering in new environments or creating 3D avatars. However, the textures provided by these methods lack diversity, missing complex styles such as cartoon styles or makeup as

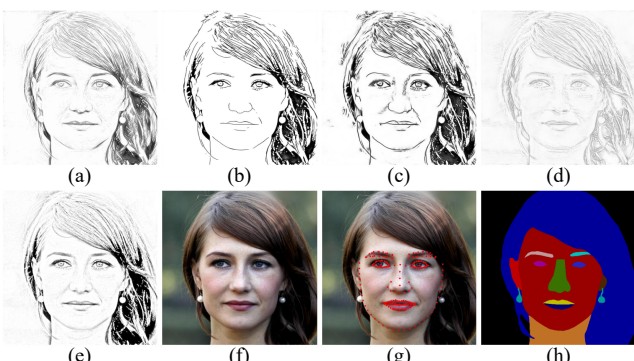

**Figure 2: Data samples of S2TD-Face. (a)-(e) are sketches in different styles generated from the original image (f). (g) represents landmarks, and (h) represents segmentation. Inputs of the pipeline include sketches (a)-(e) and (f)-(h) serve as supervisory signals.**

shown in Fig. 1, and still exhibit disparities in high-frequency details compared to textures directly derived from image UV mapping.

**Translate Sketches to Other Modalities.** Some methods [3, 22, 25, 43, 62, 64] reconstruct 3D shapes from sketches of common objects such as cups, chairs, cars, airplanes, *etc.* They typically supervise the 3D geometry based on 2D silhouettes using differential renderers [62], involving point set matching and optimization (such as chamfer distance) [22, 25], or innovation in 3D representation forms (such as Signed Distance Fields [46]) [43, 64]. These methods are usually limited to specific types of objects, and a domain gap exists when reconstructing faces. Few methods reconstruct 3D faces from facial sketches, they either specialize in sketches with frontal poses [28] or heavily depend on professional sketches with precise shading as input [61], which may not align with practical requirements. Some methods [11–13, 23, 40, 52] translate facial sketches into 2D face images, often utilizing frameworks such as Generative Adversarial Networks (GANs) [15] or Neural Radiance Fields (NeRFs) [45] to synthesize face images. Combining these sketch-to-face-image methods with image-to-3D-face methods [19, 59, 66] seems like a straightforward solution. However, the local stroke information of the original input sketch (such as dimples, wrinkles, *etc.*) is easily lost in this process, leading to reconstruction results that are not consistent with the input sketches.

## 3 METHODOLOGY

### 3.1 Preliminaries

**Data Processing.** For a given RGB face image $I \in \mathbb{R}^{H \times W \times 3}$, based on the common practices in [13, 52, 56, 57], we generate 5 types of sketches $S_{t_i} \in \mathbb{R}^{H \times W \times 3}$:

$$S_{t_i} = \Phi_{\text{sketch}}(I, t_i) \ , \tag{1}$$

where we integrate existing various sketch operations [7, 56, 57] into a single function $\Phi_{\text{sketch}}(\cdot)$. We make $\Phi_{\text{sketch}}(\cdot)$ differentiable and will also apply it to the sketch-to-geometry loss. $t_i$ represents different sketch types, $i \in [1, \cdots, 5]$, as illustrated in Fig. 2(a)-(e). Following [59], we utilize landmark detectors [59] to obtain 2D landmarks $\boldsymbol{lmk} \in \mathbb{R}^{2 \times 240}$ and employ DML-CSR [63] to generate

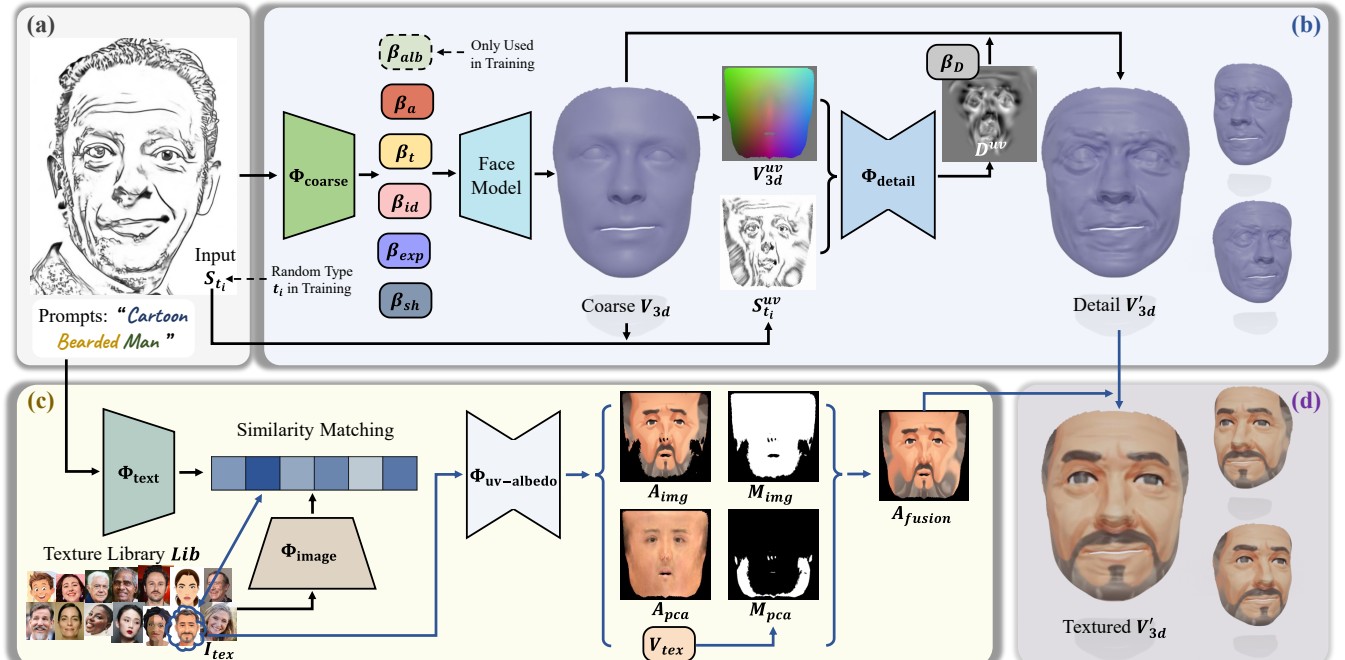

**Figure 3: Overview of our method. (a) The input of S2TD-Face: a face sketch and a text prompt. (b): The geometry reconstruction framework yields detailed 3D faces that accurately reflects the delicate features of the input sketches. (c): The texture control module seamlessly applies the controllable texture to the geometry with text prompts. (d) The output of S2TD-Face: a detailed 3D face with controllable texture.**

segmentation information $C$ for supervisory signals in geometry. These data processing methods enable S2TD-Face to acquire training data from existing abundant face datasets [32, 38, 39, 42, 44, 53], without relying on hard-to-collect 3D face scanning data or labor-intensive hand-drawn sketches. In summary, during the training process, each data sample consists of sketches $\{S_{t_i}\}$, original face image $I$, segmentation $C$, and landmarks $lmk$, as shown in Fig. 2.

**Face Model.** Based on [8, 27, 48], we define the coarse vertices and albedo of a 3D face using the following formula:

$$V_{3d} = R(\boldsymbol{\beta}_a)(\overline{V} + \boldsymbol{\beta}_{id}B_{id} + \boldsymbol{\beta}_{exp}B_{exp}) + \boldsymbol{\beta}_t \\ T_{alb} = \overline{T} + \boldsymbol{\beta}_{alb}B_{alb} \quad , \quad (2)$$

where $\overline{V}$ and $\overline{T}$ are the mean geometry and the mean albedo, respectively. $V_{3d} \in \mathbb{R}^{3 \times 35709}$ is the coarse face vertices and $T_{alb} \in \mathbb{R}^{3 \times 35709}$ is the albedo. $\boldsymbol{\beta}_{id} \in \mathbb{R}^{80}$, $\boldsymbol{\beta}_{exp} \in \mathbb{R}^{64}$ and $\boldsymbol{\beta}_{alb} \in \mathbb{R}^{80}$ are the identity geometry parameter, the expression geometry parameter and the albedo parameter, respectively. $B_{id}$, $B_{exp}$ and $B_{alb}$ are the face identity bases, the expression bases and the albedo bases, respectively. We utilize angles $\boldsymbol{\beta}_a \in \mathbb{R}^3$ (pitch, yaw, and roll) to obtain the rotation matrix $R(\boldsymbol{\beta}_a) \in \mathbb{R}^{3 \times 3}$, for the rotation of $V_{3d}$. We employ $\boldsymbol{\beta}_t \in \mathbb{R}^3$ to control the translation of $V_{3d}$. Note that $T_{alb}$ is not the final facial texture and it will not appear during the inference process of the framework. $T_{alb}$ solely assists in supervising the geometry during the training process.

**Face Attributes in UV Space.** UV mapping is a reversible 3D modeling process commonly used to project the attributes of 3D objects into the 2D image plane. We can transfer facial geometry information, facial texture information, and other attributes to UV space.

These techniques are employed in many 3D face reconstruction methods [10, 18, 20], often combined with differentiable renderers [34, 51], facial texture completion [2, 10, 18], and illumination estimation [19]. In the following section, we denote the facial attribute $X$ in UV space as $X^{uv}$.

**Camera.** Following [17, 37, 59], we utilize a camera with a fixed perspective projection for the re-projection of $V_{3d}$ into the 2D image plane, yielding $V_{2d} \in \mathbb{R}^{2 \times 35709}$.

**Illumination Model.** Based on [17, 19], we employ Spherical Harmonics (SH) [50] to predict the shading information:

$$S(\boldsymbol{\beta}_{sh}, A, N) = A \odot \sum_{k=1}^{9} \boldsymbol{\beta}_{sh}^k \Psi_k(N) \quad , \quad (3)$$

where $\odot$ denotes the Hadamard product, $N$ is the surface normal of $V_{3d}$, $\Psi : \mathbb{R}^3 \rightarrow \mathbb{R}$ is the SH basis function and $\boldsymbol{\beta}_{sh}^k \in \mathbb{R}^3$ is the corresponding SH parameter, $k \in [1, \cdots, 9]$. $A$ represents the albedo information, which could be set as $T_{alb}$ to calculate the shaded texture. Following [19, 37], we also set $A$ to a fixed gray value $A_{gray}$ to display the geometry shading.

**Detail Reconstruction.** The coarse geometry $V_{3d}$ based on 3DMMs can not capture the high-frequency details of a 3D face. To address this, we perform detail reconstruction based on [19, 37], which is achieved by computing a displacement map:

$$V'_{3d}{}^{uv} = V_{3d}{}^{uv} + \boldsymbol{\beta}_D D^{uv} \odot N^{uv} \quad , \quad (4)$$

where $D^{uv} \in \mathbb{R}^{256 \times 256}$ represents the detail displacement map in UV space. $V_{3d}{}^{uv} \in \mathbb{R}^{256 \times 256 \times 3}$ and $V'_{3d}{}^{uv} \in \mathbb{R}^{256 \times 256 \times 3}$ denote

the coarse geometry and detail geometry in UV space, respectively. $N^{uv} \in \mathbb{R}^{256 \times 256 \times 3}$ represents the surface normal corresponding to $V_{3d}{}^{uv}$. $\beta_D \in \mathbb{R}^+$ is used to control the magnitude of the displacement map $D^{uv}$. We denote the surface normal of the detail geometry $V'_{3d}$ as $N'$.

**Rendering.** Based on [19, 34, 51], we construct a differentiable renderer $\Phi_{\text{render}}(\cdot)$ using the fixed camera, which could yield the following results:

$$
\begin{aligned}
I^a &= \Phi_{\text{render}}(V_{3d}, S(\beta_{sh}, T_{alb}, N)) \\
I^b &= \Phi_{\text{render}}(V_{3d}, S(\beta_{sh}, T_{alb}, N')) \\
I^c &= \Phi_{\text{render}}(V_{3d}, S(\beta_{sh}, A_{gray}, N)) \\
I^d &= \Phi_{\text{render}}(V_{3d}, S(\beta_{sh}, A_{gray}, N'))
\end{aligned}
\quad, \tag{5}
$$

where the input of the differentiable renderer $\Phi_{\text{render}}(\cdot)$ includes coarse geometry and shading information, achieving detailed rendering effects through the refinement of the normal map in the shading information. The rendering results $I^a$, $I^b$, $I^c$, and $I^d$ represent the coarse texture, detail texture, coarse geometry shading, and detail geometry shading, respectively, which are used in sketch-to-geometry loss, as shown in Fig. 4.

## 3.2 Geometry Reconstruction Framework

We aim to reconstruct detailed geometry consistent with the delicate features of the input sketch. The sketch-to-3D-face process of S2TD-Face is divided into two stages: coarse geometry reconstruction and detailed geometry reconstruction, as shown in Fig. 3(b).

**Coarse Geometry Reconstruction.** During the training process, for each data sample, we randomly select a sketch $S_{t_i}$ of type $t_i$ as input. We employ ResNet-50 [29] as the backbone $\Phi_{\text{coarse}}$ to predict parameters $\beta_a$, $\beta_t$, $\beta_{id}$, $\beta_{exp}$, $\beta_{sh}$, and $\beta_{alb}$. These parameters are processed by the face model [8, 48] to generate coarse geometry $V_{3d}$ and the PCA albedo $T_{alb}$, as described in Eqn. 2. $T_{alb}$ is used for photometric loss $\mathcal{L}_{\text{pho}}$, perception loss $\mathcal{L}_{\text{per}}$, and sketch-to-geometry loss $\mathcal{L}_{\text{sketch}}$. Note that during the inference, there are no restrictions on the sketch types, and neither $T_{alb}$ nor $\beta_{alb}$ are involved. Additionally, $\beta_{sh}$ for controlling light can vary as shown in Fig. 1. We utilize $V_{3d}$ to map the sketch image $S_{t_i}$ to UV space, resulting in $S_{t_i}^{uv}$. $S_{t_i}^{uv}$. The UV space representation $V_{3d}^{uv}$ of $V_{3d}$ will jointly serve as the input for reconstructing the detailed geometry.

**Detailed Geometry Reconstruction.** Using $V_{3d}^{uv}$ and $S_{t_i}^{uv}$ as input, we employ a pix2pix network [30] $\Phi_{\text{detail}}$ to predict the displacement map $D^{uv}$ for reconstructing the detailed geometry $V'_{3d}{}^{uv}$ in Eqn. 4. Throughout this process, $\beta_D$ serves as a learnable parameter controlling the magnitude of $D^{uv}$, which is fixed during inference.

## 3.3 Training Strategies

To train $\Phi_{\text{coarse}}$ and $\Phi_{\text{detail}}$ in S2TD-Face, we employ the following training methods and supervision loss functions.

**Various Sketch Types.** The facial sketch types are diverse, with some containing realistic shading information, while others only consist of simple lines. To ensure the robustness of $\Phi_{\text{coarse}}$ and $\Phi_{\text{detail}}$ across different sketches, randomization is applied to all operations involving sketch type selection during the training process. Specifically, the training input employs a random sketch type $t_i$, as

shown in Fig. 3(b), and in the sketch-to-geometry loss $\mathcal{L}_{\text{sketch}}$, the type $t_j$ is also randomly selected, as shown in Fig. 4. These strategies ensures that $\Phi_{\text{coarse}}$ and $\Phi_{\text{detail}}$ possess strong adaptability to different sketch types.

**Sketch-to-geometry Loss.** Existing supervision methods fail to accurately capture the local details of sketches (such as dimples, wrinkles, *etc.*) and reflect them onto the geometry deformation. To address this, we propose a novel sketch-to-geometry loss $\mathcal{L}_{\text{sketch}}$ to ensure the geometry $V_{3d}$ and $V'_{3d}$ fidelity to the sketch input and supervise $\Phi_{\text{coarse}}$ and $\Phi_{\text{detail}}$ robustly across different sketch types, as shown in Fig. 4. Based on the rendering results $I^a$, $I^b$, $I^c$, and $I^d$, we further utilize $\Phi_{\text{sketch}}(\cdot)$ to generate the corresponding sketches $S_{t_j}^a$, $S_{t_j}^b$, $S_{t_j}^c$, and $S_{t_j}^d$:

$$
S_{t_j}^n = \Phi_{\text{sketch}}(I^n, t_j), \text{ for } n \in \{a, b, c, d\} \quad, \tag{6}
$$

where type $t_j$ is randomly selected. Since $\Phi_{\text{coarse}}$, $\Phi_{\text{detail}}$, $\Phi_{\text{render}}$, and $\Phi_{\text{sketch}}$ are all differentiable, and we have the sketch ground truth $S_{t_j}$ corresponding to the type $t_j$, we could compare the differences between $\{S_{t_j}^n | n \in \{a, b, c, d\}\}$ and $S_{t_j}$ by using photometric loss and perception loss:

$$
\mathcal{L}_{\text{sketch}} = \lambda_1 \underbrace{\sum_{n \in \{a,b,c,d\}} \left\| M^n - M \right\|_2}_{\text{sketch}-\text{photometric}}
$$
$$
+ \lambda_2 \underbrace{\sum_{n \in \{a,b,c,d\}} (1 - \frac{< \Phi_{\text{per}}(M^n), \Phi_{\text{per}}(M) >}{||\Phi_{\text{per}}(M^n)||_2 \cdot ||\Phi_{\text{per}}(M)||_2})}_{\text{sketch}-\text{perception}}, \tag{7}
$$

where $\mathcal{L}_{\text{sketch}}$ contains two error parts: photometric error and perception error. The former computes L2-norm error, while the latter computes the cosine distance. $\lambda_1$ and $\lambda_2$ are the corresponding weights. $\Phi_{\text{per}}(\cdot)$ is a face recognition network from [17], used to extract features from the input, and $< \cdot, \cdot >$ denotes the vector inner product. $M^n$ and $M$ respectively represent the mask-filtered results of $\{S_{t_j}^n | n \in \{a, b, c, d\}\}$ and $S_{t_j}$, i.e. $M^n = M_C \odot M_{render} \odot S_{t_j}^n$, for $n \in \{a, b, c, d\}$ and $M = M_C \odot M_{render} \odot S_{t_j}$. $M_C$ and $M_{render}$ respectively represent the masks obtained by segmentation information $C$ and $\Phi_{\text{render}}$, as shown in Fig. 4. Combining with the mask-filtered results can eliminate interference caused by occlusions and focus on the rendering object.

**Photometric Loss and Perception Loss for $I^b$.** To enhance the robustness of the training process, we supervise the detail texture rendering result $I^b$ in Eqn. 5 similar to [19, 37]. Note that this process operates at the rendering image level $I^b$, while $\mathcal{L}_{\text{sketch}}$ operates at the rendering sketch level. The photometric loss $\mathcal{L}_{\text{pho}}$ and perception loss $\mathcal{L}_{\text{per}}$ used are defined as follows:

$$
\mathcal{L}_{\text{pho}} = \left\| (MI^b - MI) \right\|_2 \quad, \tag{8}
$$

$$
\mathcal{L}_{\text{per}} = 1 - \frac{< \Phi_{\text{per}}(MI^b), \Phi_{\text{per}}(MI) >}{\left\| \Phi_{\text{per}}(MI^b) \right\|_2 \cdot \left\| \Phi_{\text{per}}(MI) \right\|_2}, \tag{9}
$$

where $MI^b = M_C \odot M_{render} \odot I^b$ and $MI = M_C \odot M_{render} \odot I$. Similar to the operations in $\mathcal{L}_{\text{sketch}}$, $\Phi_{\text{per}}(\cdot)$ is a face recognition network [17] and $< \cdot, \cdot >$ is the vector inner product. We emphasize

**Figure 4: Overview of sketch-to-geometry loss.** $\mathcal{L}_{\text{sketch}}$ **compares the predicted sketches** $\{S_{t_j}^a, S_{t_j}^b, S_{t_j}^c, S_{t_j}^d\}$ **with the ground truth** $S_{t_j}$ **to supervise the geometry deformation, obtaining detailed geometry consistent with the delicate features of the input.**

again that in $\mathcal{L}_{\text{sketch}}$, $\mathcal{L}_{\text{pho}}$, and $\mathcal{L}_{\text{per}}$, the texture of $I^a$ or $I^b$ is derived from $T_{alb}$, aims to supervise the geometry. $T_{alb}$ is not the final texture and will not appear in the inference process.

**Landmark Loss.** We employ landmark loss to compare the predicted 2D landmarks $lmk'$ from $V_{2d}$ with the ground truth $lmk$ obtained by [59], adopting the dynamic landmark marching [65] to address the non-correspondence between 2D and 3D cheek contour caused by pose variations. The landmark loss $\mathcal{L}_{\text{lmk}}$ is defined as:

$$\mathcal{L}_{\text{lmk}} = \sum_{i=1}^{240} \left\| lmk'_i - lmk_i \right\|_2. \tag{10}$$

**Part Re-projection Distance Loss.** Since $\mathcal{L}_{\text{lmk}}$ can only operate on sparse vertices in $V_{2d}$, we further utilize Part Re-projection Distance Loss (PRDL) [59] $\mathcal{L}_{\text{prdl}}$ to supervise $V_{2d}$. $\mathcal{L}_{\text{prdl}}$ leverages the precise 2D part silhouettes provided by segmentation $C$ to constrain the predicted geometry of facial features:

$$\mathcal{L}_{\text{prdl}} = \sum_{p \in P} \lambda_p \left\| \Gamma_p(V_{2d}) - \Gamma_p(C) \right\|_2, \tag{11}$$

where $P$ represents the set of facial components, $P$ = {left_eye, right_eye, left_eyebrow, right_eyebrow, up_lip, down_lip, nose, skin}. $\Gamma_p(V_{2d})$ and $\Gamma_p(C)$ respectively denote the shape descriptors of PRDL defined for prediction and target in [59]. $\lambda_p$ represents the weight of each part $p$. During training, specific parts $p$ of samples may be occluded or invisible, we set $\lambda_p = 0$ for parts $p$ in these samples and $\lambda_p = 1$ for the rest parts.

**Overall Losses.** In summary, we minimize the total loss $\mathcal{L}$ to optimize the geometry reconstruction frameworks $\Phi_{\text{coarse}}$ and $\Phi_{\text{detail}}$:

$$\begin{aligned} \mathcal{L} = \ &\lambda_{\text{sketch}} \mathcal{L}_{\text{sketch}} + \lambda_{\text{pho}} \mathcal{L}_{\text{pho}} + \lambda_{\text{per}} \mathcal{L}_{\text{per}} \\ &+ \lambda_{\text{lmk}} \mathcal{L}_{\text{lmk}} + \lambda_{\text{prdl}} \mathcal{L}_{\text{prdl}} + \lambda_{\text{reg}} \mathcal{L}_{\text{reg}}, \end{aligned} \tag{12}$$

where $\mathcal{L}_{\text{reg}}$ is the regularization loss for parameters $\beta$. $\lambda_{\text{sketch}} = 1$, $\lambda_1 = 1.33$, $\lambda_2 = 0.1$, $\lambda_{\text{pho}} = 0.57$, $\lambda_{\text{per}} = 0.1$, $\lambda_{\text{lmk}} = 1.6e - 3$, $\lambda_{\text{prdl}} = 8e - 4$, and $\lambda_{\text{reg}} = 3e - 4$ are the corresponding weights. $\mathcal{L}_{\text{lmk}}$ and $\mathcal{L}_{\text{prdl}}$ are normalized by $H \times W$.

**Training Details.** We firstly train $\Phi_{\text{coarse}}$, then freeze $\Phi_{\text{coarse}}$ to train $\Phi_{\text{detail}}$ and finally train $\Phi_{\text{coarse}}$ and $\Phi_{\text{detail}}$ together. Therefore, during the first training stage when using $\mathcal{L}_{\text{sketch}}$, $S_{t_j}^b$ and $S_{t_j}^d$ are not used, and $I^b$ in $\mathcal{L}_{\text{pho}}$ and $\mathcal{L}_{\text{per}}$ is replaced by $I^a$.

## 3.4 Texture Control Module

We aim to design a texture control module for S2TD-Face that can select appropriate samples from a given texture library based on input text prompts, obtain textures in UV space, and seamlessly map them onto the geometry $V'_{3d}$. As illustrated in the Fig. 3(c), when the input prompt $Text$ is 'Cartoon Beard Man', we use $\Phi_{\text{image}}$ and $\Phi_{\text{text}}$ from CLIP [49] to encode the face images $I^i_{Lib}$ from the known texture library $Lib$ and the input text $Text$:

$$\begin{aligned} F_i^I &= \Phi_{\text{image}}(I^i_{Lib}), i = 1, 2, \cdots, |Lib| \\ F^T &= \Phi_{\text{text}}(Text) \end{aligned}, \tag{13}$$

where $F_i^I$ and $F^T$ are the image encoding features and the text encoding features, respectively. $|Lib|$ is the image number in the given texture library $Lib$. In the text-to-image matching process, each $F_i^I$ is compared to $F^T$ to compute similarity, and S2TD-Face can either select the image with maximum similarity to the input text or randomly choose one from the top-k similar images. We denote the final matching result as $I_{tex}$.

The role of $\Phi_{\text{uv}-\text{albedo}}$ in the texture control module is to transform the texture of the face image into UV spcae that are compatible with $V'_{3d}$. We input the text-image matching result $I_{tex}$ to $\Phi_{\text{uv}-\text{albedo}}$ to get the desired texture information in UV space. Specifically, $\Phi_{\text{uv}-\text{albedo}}$ is based on the state-of-the-art monocular 3D face reconstruction method [59]. $\Phi_{\text{uv}-\text{albedo}}$ firstly estimates the 3DMMs [8, 48] shape $V_{tex}$ and the PCA albedo $A_{pca}$ from $I_{tex}$, and then utilizes the shape information $V_{tex}$ to map the input image $I_{tex}$ into UV space, obtaining $A_{img}$, as shown in the Fig. 3(c). Due to the pose influence of $V_{tex}$, some facial areas of $I_{tex}$ are invisible and the UV-texture information $A_{img}$ may not cover the entire facial surface. Therefore, $\Phi_{\text{uv}-\text{albedo}}$ calculates the invisible areas according to $V_{tex}$ and complete UV-texture using $A_{pca}$, finally resulting in the fusion texture $A_{fusion}$:

$$A_{fusion} = M_{img} \odot A_{img} + M_{pca} \odot A_{pca}, \tag{14}$$

where $M_{img}$ is a mask computed by the differentiable renderer $\Phi_{\text{render}}$ with the help of $V_{tex}$, which represents the visible regions of the reconstructed shape $V_{tex}$ that consistent with the input texture mapping $A_{img}$. $M_{pca} = 1 - M_{img}$ indicates the regions that require further complement by the predicted 3DMMs PCA albedo $A_{pca}$. To reduce visual differences at the fusion boundaries, we apply median filtering [7] to $M_{img}$. The texture control module finally applies $A_{fusion}$ onto the geometry $V'_{3d}$ through UV mapping, resulting in a detailed and textured 3D face, as shown in the Fig. 3(d).

**Table 1: Quantitative comparison on Sketch-REALY benchmark. We transform the test samples from REALY [9] into two types of sketches: 'Shading' (realistic shaded sketches) and 'Line' (sparse line sketches), as shown in Fig. 5, and perform quantitative comparison respectively. Lower values indicate better results. The best and runner-up are highlighted in bold and underlined, respectively. We also investigate the effect of removing the sketch-to-geometry loss $\mathcal{L}_{sketch}$ (denoted as 'Ours (w/o $\mathcal{L}_{sketch}$)') for ablation study.**

| Types | Methods | Frontal-view (mm) ↓ | | | | | Side-view (mm) ↓ | | | | |
|---|---|---|---|---|---|---|---|---|---|---|---|
| | | Nose avg.± std. | Mouth avg.± std. | Forehead avg.± std. | Cheek avg.± std. | avg. | Nose avg.± std. | Mouth avg.± std. | Forehead avg.± std. | Cheek avg.± std. | avg. |
| Shading | PRNet [20]† | 2.047±0.498 | 1.750±0.569 | 2.400±0.586 | 1.896±0.694 | 2.023 | 2.027±0.507 | 1.880±0.591 | 2.525±0.643 | 2.093±0.757 | 2.131 |
| | MGCNet [54]† | 1.711±0.422 | 1.617±0.552 | 2.194±0.567 | 1.609±0.588 | 1.783 | 1.685±0.438 | 1.555±0.511 | 2.189±0.560 | 1.656±0.597 | 1.771 |
| | Deep3D[17]† | 1.781±0.430 | 1.714±0.592 | 2.124±0.482 | 1.274±0.461 | 1.723 | 1.658±0.350 | 1.830±0.663 | 2.147±0.502 | 1.284±0.466 | 1.730 |
| | 3DDFA-V2 [26]† | 1.866±0.498 | 1.722±0.503 | 2.509±0.687 | 1.956±0.709 | 2.013 | 1.856±0.489 | 1.724±0.522 | 2.535±0.660 | 1.993±0.723 | 2.027 |
| | HRN [37]† | 1.723±0.435 | 1.878±0.623 | 2.202±0.497 | 1.246±0.424 | 1.762 | 1.647±0.369 | 1.957±0.693 | 2.245±0.515 | 1.269±0.420 | 1.779 |
| | DECA [19]† | 1.830±0.405 | 2.475±0.793 | 2.420±0.598 | 1.600±0.597 | 2.081 | 1.858±0.428 | 2.542±0.836 | 2.448±0.610 | 1.628±0.607 | 2.119 |
| | DeepSketch2Face [28] | 3.896±0.774 | 2.808±1.392 | 5.091±0.899 | 6.450±0.972 | 4.561 | 3.950±0.810 | 3.250±1.669 | 5.489±1.069 | 6.746±1.038 | 4.859 |
| | Ours (w/o $\mathcal{L}_{sketch}$) | **1.621±0.323** | 1.454±0.487 | 2.021±0.492 | 1.288±0.378 | 1.596 | 1.594±0.317 | 1.482±0.509 | 2.041±0.565 | 1.299±0.385 | 1.604 |
| | **Ours** | 1.630±0.348 | **1.324±0.412** | **1.986±0.418** | **1.191±0.343** | **1.533** | **1.559±0.329** | **1.357±0.469** | **1.960±0.471** | **1.149±0.336** | **1.506** |
| Line | PRNet [20]† | 2.166±0.553 | 2.127±0.648 | 2.714±0.787 | 2.164±0.798 | 2.293 | 2.138±0.552 | 2.243±0.821 | 3.071±0.985 | 2.422±0.894 | 2.468 |
| | MGCNet [54]† | 2.114±0.632 | 2.257±0.851 | 2.881±0.946 | 1.714±0.630 | 2.241 | 2.039±0.532 | 2.019±0.730 | 2.840±0.994 | 1.800±0.689 | 2.175 |
| | Deep3D[17]† | 2.230±0.513 | 1.865±0.646 | 2.290±0.550 | 1.487±0.542 | 1.968 | 1.975±0.483 | 1.876±0.650 | 2.354±0.605 | 1.475±0.549 | 1.920 |
| | 3DDFA-V2 [26]† | 1.965±0.561 | 2.045±0.685 | 2.632±0.798 | 1.931±0.752 | 2.143 | 1.968±0.551 | 2.056±0.672 | 2.681±0.838 | 1.976±0.805 | 2.170 |
| | HRN [37]† | 2.152±0.553 | 1.974±0.654 | 2.579±0.720 | 1.614±0.692 | 2.080 | 2.057±0.547 | 2.089±0.736 | 2.669±0.839 | 1.580±0.609 | 2.099 |
| | DECA [19]† | 2.121±0.490 | 2.598±0.914 | 2.703±0.606 | 1.641±0.573 | 2.266 | 2.071±0.482 | 2.559±0.947 | 2.757±0.696 | 1.630±0.573 | 2.254 |
| | DeepSketch2Face [28] | 3.359±0.653 | 2.483±0.595 | 4.835±0.994 | 5.464±1.074 | 4.035 | 3.726±0.895 | 2.701±0.717 | 5.150±1.037 | 6.124±1.086 | 4.425 |
| | Ours (w/o $\mathcal{L}_{sketch}$) | **1.688±0.359** | 1.755±0.640 | 2.288±0.553 | 1.477±0.383 | 1.802 | 1.675±0.352 | 1.798±0.594 | 2.316±0.618 | 1.495±0.397 | 1.821 |
| | **Ours** | 1.692±0.366 | **1.524±0.505** | **2.131±0.510** | **1.344±0.385** | **1.673** | **1.627±0.350** | **1.556±0.476** | **2.227±0.570** | **1.352±0.377** | **1.690** |

† There are two ways to reconstruct 3D faces from sketches based on existing SOTA methods [17, 19, 20, 26, 37, 54]: firstly translating 2D sketches to face images [52] and subsequently reconstructing 3D faces or directly using sketches as input. **For fairness, methods marked with † represent the best results obtained from these two ways.**

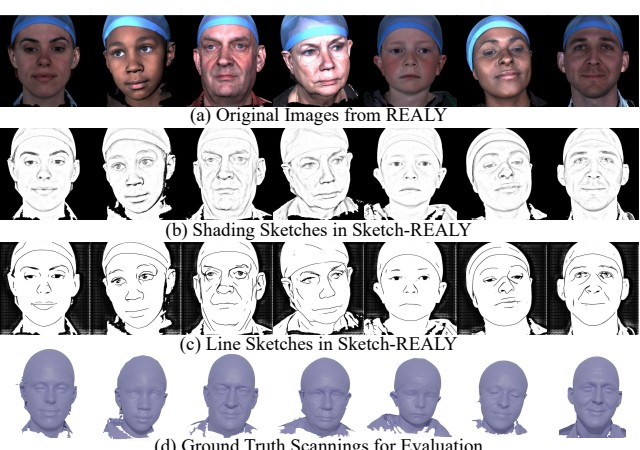

(a) Original Images from REALY

(b) Shading Sketches in Sketch-REALY

(c) Line Sketches in Sketch-REALY

(d) Ground Truth Scannings for Evaluation

**Figure 5: The test samples (7/100) of Sketch-REALY. (a): The original test images from REALY [9]. (b) and (c): The 2 styles (Shading and Line) of the test images in Sketch-REALY. (d): The face scanning for geometry evaluation in Sketch-REALY.**

## 4 EXPERIMENTS

### 4.1 Experimental Settings

**Implementation Details.** We implement S2TD-Face based on PyTorch [47]. The input sketches are resized into 224×224. $A_{gray} = (127, 127, 127)$. We use Adam [33] as the optimizer with an initial learning rate of $1e − 4$. $B_{id}$ and $B_{alb}$ are from BFM2009 [48] and $B_{exp}$ is from FaceWarehouse [8].

**Data.** We utilize face images from publicly available datasets, including CelebA [42], 300W [53], RAF [38, 39], and DAD-3DHeads [44], which are commonly used in 3D face reconstruction tasks. We employ [66] for face pose augmentation and [59] for face expression augmentation. As a result, we obtain about $600K$ face images for training. $\Phi_{sketch}$ is based on [7, 56, 57] and each face image is processed by $\Phi_{sketch}$ to obtain 5 different styles of sketches as input to the framework (resulting in $5 \times 600K$ sketches). The images in the texture library *Lib* of the texture control module are sourced from FFHQ [32] and online collections, totaling about 1000 images.

### 4.2 Metrics

**Sketch-REALY.** The REALY benchmark [9] comprises 100 precise 3D face scanning data (as shown in Fig. 5 (d)) from LYHM [16], which are from different identities and include accurate landmarks, region masks, and topology-consistent meshes. During the evaluation, REALY initially aligns the prediction and ground truth using landmarks. It subsequently divides the reconstructed results into 4 parts (nose, mouth, forehead, and cheek) using region masks. Finally, it utilizes the ICP algorithm [1] for precise registration between prediction and ground truth and computes the corresponding average Normalized Mean Square Error (NMSE) for different face regions. The REALY test samples are divided into 2 parts, consisting of 100 frontal-view images and 400 side-view images. REALY has served as the benchmark for geometric evaluation by most state-of-the-art methods [10, 18, 59]. We propose a new Sketch-REALY benchmark based on REALY [9] to tailor it for sketch-to-3D-face reconstruction tasks, highlighting the performance of geometry

**Figure 6: More visualization results of our method (S2TD-Face). S2TD-Face can reconstruct high-fidelity geometry from face sketches of different styles and generate controllable textures spanning cartoon, sculptural, and realistic facial styles guided by text prompts. The results can also be re-lighted for broader applications.**

reconstruction from sketches. Specifically, we use $\Phi_{\text{sketch}}$ to process the REALY test images, generating 2 different types of sketches. The former retains the shading information from the original images, resembling realistic grayscale images (denoted as 'Shading'), while the latter only consists of sparse lines (denoted as 'Line'), as shown in Fig. 5 (b) and (c). We conduct the geometry evaluation on the 3D prediction of these two types of sketches, thereby establishing the Sketch-REALY benchmark.

**SSIM and PSNR.** Structural Similarity Index Measure (SSIM) [58] and Peak Signal to Noise Ratio (PSNR) are two standard metrics used to measure the similarity between images. SSIM considers the brightness, contrast, and structural information of the images, with values ranging from 0 to 1, where higher values indicate greater similarity. PSNR evaluates the similarity between images by computing the mean squared error between them. PSNR typically ranges from 0 to infinity and is measured in decibels (dB). Higher PSNR values indicate smaller differences between the images, reflecting higher similarity. In our ablation study, we utilize SSIM and PSNR to quantify the differences between the coarse or detail geometry shading sketches ($S_{t_j}^c$ or $S_{t_j}^d$) and the ground truth $S_{t_j}$, thereby quantitatively evaluating the impact of $\Phi_{\text{detail}}$ and $\mathcal{L}_{\text{sketch}}$ on visual quality.

### 4.3 Quantitative Comparison

Based on the Sketch-REALY benchmark, we comprehensively evaluated our methods with state-of-the-art approaches, including MGCNet [54], PRNet [20], HRN [37], Deep3D [17], 3DDFA-V2 [26], DECA [19], and DeepSketch2Face [28]. DeepSketch2Face [28] is a method tailored for sketch-to-3D-face reconstruction tasks,

whereas [17, 19, 20, 26, 37, 54] are commonly used for reconstructing RGB face images. There are two ways to reconstruct 3D faces from sketches using these methods: firstly translating 2D sketches to face images [52] and subsequently reconstructing 3D faces or directly inputting sketches. To ensure fairness, we present the best results of these both ways for [17, 19, 20, 26, 37, 54]. The evaluation results on Sketch-REALY are shown in Tab.1. Tab.1 indicates that our method achieved the best results on both shading sketches (1.533$mm$ in frontal-view and 1.506$mm$ in side-view) and hard test cases with sparse line sketches (1.673$mm$ in frontal-view and 1.690$mm$ in side-view), surpassing the second-best method by a considerable margin, indicating that our method exhibits superior robustness to the type and pose of the input facial sketch.

### 4.4 Qualitative Comparison

Similar to Fig. 1, Fig. 6 further illustrates the visualization results of S2TD-Face. S2TD-Face is capable of reconstructing high-fidelity 3D faces consistent with the input sketch details from various styles. It can provide controllable textures based on text prompts, ranging from cartoon-style, sculptural style to realistic facial style. We also compared the reconstruction results of our method with those of DeepSketch2Face [28], Deep3D [17], and HRN [37], as shown in the Fig. 7. DeepSketch2Face [28] is limited to reconstructing 3D exaggerating faces from sketches with a frontal pose and cannot handle side views or adapt to various sketch styles. Deep3D [17] is only capable of reconstructing coarse geometry. While HRN [37] can capture high-frequency facial details, it may encounter failures in certain samples. The qualitative comparison indicates that S2TD-Face is capable of handling various styles and poses of

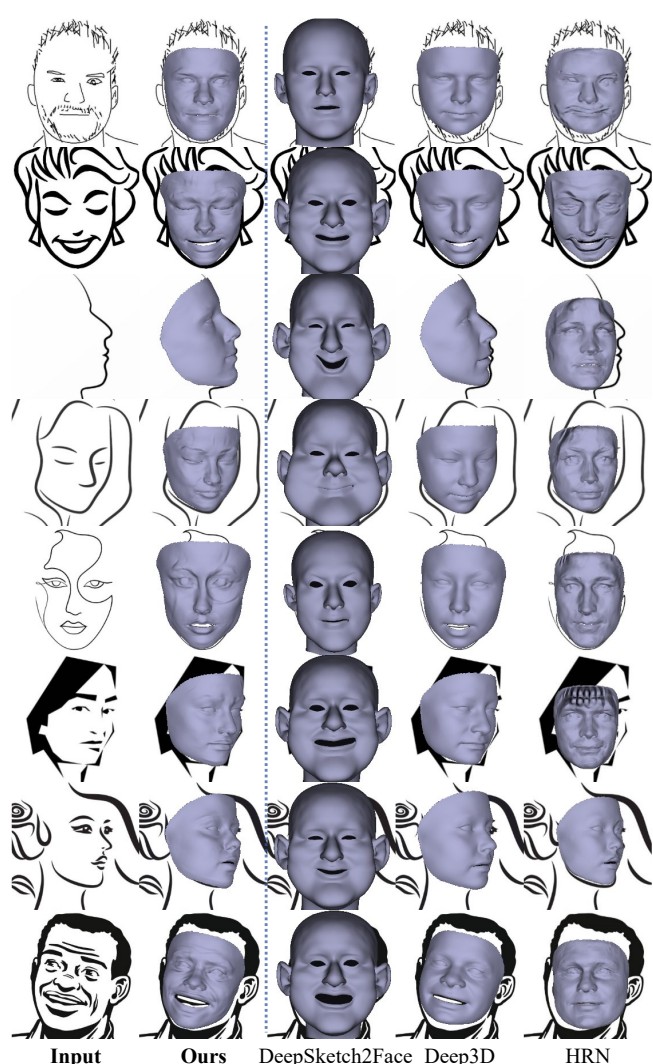

**Input**     **Ours**     DeepSketch2Face    Deep3D     HRN

Figure 7: Qualitative comparison with the other methods. Our method (S2TD-Face) achieves the best results that consistent with the input sketch details.

facial sketches and achieves the best results consistent with the input sketch details. Note that these methods [17, 28, 37] all lack the ability to control the texture of the reconstructed 3D faces, while our S2TD-Face uniquely offers this capability.

## 4.5 Ablation Study

**Impact of $\mathcal{L}_{\text{sketch}}$ on Geometry.** We investigate the effect of sketch-to-geometry loss $\mathcal{L}_{\text{sketch}}$ for supervising geometry deformation. As shown in Tab. 1, based on our proposed geometry reconstruction framework and Sketch-REALY benchmark, we present results for both when the framework is applied independently (denoted as 'Ours (w/o $\mathcal{L}_{\text{sketch}}$)') and when combined with $\mathcal{L}_{\text{sketch}}$ (denoted as 'Ours'). The former indicates our geometry reconstruction framework performs superior to existing state-of-the-art methods across most cases. The latter further shows that incorporating

Table 2: Ablation study for the impact of $\mathcal{L}_{\text{sketch}}$ and $\Phi_{\text{detail}}$ on visual quality. Higher values indicate better results and the best is highlighted in bold.

| $\Phi_{\text{coarse}}$ | $\mathcal{L}_{\text{sketch}}$ | $\Phi_{\text{detail}}$ | SSIM ↑ | PSNR ↑ |
|:---:|:---:|:---:|:---:|:---:|
| ✓ | ✗ | ✗ | 0.764 | 25.11 |
| ✓ | ✗ | ✓ | 0.776 | 25.22 |
| ✓ | ✓ | ✗ | 0.789 | 26.27 |
| ✓ | ✓ | ✓ | **0.799** | **26.51** |

$\mathcal{L}_{\text{sketch}}$ contributes to improved geometry deformation. The combination of $\mathcal{L}_{\text{sketch}}$ with our geometry reconstruction framework further refines the accuracy of the reconstructed geometry.

**Impact of $\mathcal{L}_{\text{sketch}}$ and $\Phi_{\text{detail}}$ on Visual Quality.** We use SSIM and PSNR to investigate the impact of $\mathcal{L}_{\text{sketch}}$ and $\Phi_{\text{detail}}$ on visual quality. Utilizing rendering techniques [34, 51] and sketch extraction methods [7, 56, 57], we can acquire coarse or detailed geometry shading sketches ($S_{t_j}^c$ or $S_{t_j}^d$) for the predicted results. These sketches are subsequently compared with the ground truth $S_{t_j}$ to compute the SSIM and PSNR scores. The test images are sourced from [9]. Quantitative results are shown in Tab.2. When neither $\mathcal{L}_{\text{sketch}}$ nor $\Phi_{\text{detail}}$ is involved, relying solely on $\Phi_{\text{coarse}}$ for reconstruction leads to poorer visual quality. Combining $\Phi_{\text{coarse}}$ with either $\mathcal{L}_{\text{sketch}}$ or $\Phi_{\text{detail}}$ individually results in improved visual quality, while employing $\mathcal{L}_{\text{sketch}}$ and $\Phi_{\text{detail}}$ together yields the best results. Note that comparing the second and third rows of Tab. 2, the combination of $\mathcal{L}_{\text{sketch}}$ with $\Phi_{\text{coarse}}$ even outperforms the combination of $\Phi_{\text{detail}}$ with $\Phi_{\text{coarse}}$, further indicating the effectiveness of our proposed sketch-to-geometry loss $\mathcal{L}_{\text{sketch}}$ in faithfully capturing the geometric information of the input sketch.

## 4.6 Limitations

We summarize two limitations of our methods. Firstly, although Fig.1 and Fig.6 show that S2TD-Face can generate controllable textures based on text prompts, these textures are all obtained from existing facial images through UV-mapping and UV-texture-completion techniques. Low-frequency 3DMMs PCA albedo information is employed through the completion process, sometimes leading to noticeable visual differences at the fusion boundaries. Secondly, while our method can reconstruct high-fidelity geometry from face sketches of various styles, the geometry is derived from 3DMMs, resulting in the overall appearance still resembling real human faces when dealing with the animation styles.

## 5 CONCLUSIONS

This paper proposes a method tailored to the sketch-to-3D-face task, named S2TD-Face. S2TD-Face is capable of reconstructing high-fidelity topology-consistent detailed geometry from face sketches of diverse styles. It enables the controllable textures of 3D faces spanning cartoon, sculptural, and realistic facial styles based on text prompts. The contributions include an effective geometry reconstruction framework, a novel sketch-to-geometry loss for guiding geometry deformation, and a novel texture module for texture control based on text prompts. Extensive experiments show that the outstanding performance of our method surpasses existing state-of-the-art methods. The code will be publicly available.

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
