# OpenReview forum: "S2TD-Face: Reconstruct a Detailed 3D Face with Controllable Texture from a Single Sketch"
_acmmm.org/ACMMM/2024/Conference — MM2024 Poster_

### Official Review · Reviewer_Z3Zb · 2024-05-13

**Rating:** 3
**Confidence:** 3

**Summary:**

The authors propose to reconstruct 3D detailed faces based on sketches and textual prompts. To enhance the high-fidelity geometry, a two-stage reconstruction is used to generate coarse 3d and detailed 3d respectively. In order to select a suitable texture, the input prompt text is used to select the most similar texture from the texture library based on the matching of features. Finally, textures and detailed geometries are blended to obtain a controlled 3D face. Experiments are conducted on sharing and line sketches in Sketch-REALY to verify the effectiveness.

**Strengths:**

+ The idea is intuitive to combine a 3D geometry and prompt-driven texture for the controlled final 3D reconstruction.
+ The motivation for using sketch-to-geometry loss to directly generate 3D geometry directly from sketch is clear.
+ The paper is well-written, making it easy to follow.
+ The quantitative results for 3D face reconstruction from sketch seem promising. But unsure whether customized prompts are used for each sketch.

**Limitations:**

1. Each technology component draws on existing methodologies. It is not clear what is new in the proposed method.
2. The control using prompts is not strong enough to reflect the final 3D face. What if the selected texture is far away from the generated 3D geometry? If the prompt is not accurate (boy->girl), what is the generated 3D face? The authors should introduce the pool of all input prompts and explore the impact of using different prompts in different granularity for controlling the 3D face.
3. How to ensure that landmarks are well detected in large poses or simplified sketches where key components (such as the second example face in Figure 1) are not even visible.
4. Experiments should be performed on at least one sketch benchmark other than the sketch-REALY benchmark. Many hyperparameters need to be adjusted and which ones are sensitive to the final result? Any suggestions if the proposed method is applied to different training data sets?
5. The impact of sketches in different granularity on 3D generation should also be discussed. What is the potential of the proposed method to handle simplified sketches?

More comments:

a) Why don't the authors train coarse geometries and detailed end-to-end training? Is it unstable to train the two together?

b) The authors should report the computational cost of the module (b-d) in Figure 3 especially the texture control module which uses a texture library of 1000 images for matching. What were the main considerations in building this library?

c) Since the 3D geometry construction module is based on 3DMM in the proposed framework? Have the authors tried neural radiation fields as an alternative?

d) What are the differences and connections between the proposed method and these two related works?

- [1] Gao, L., Liu, F.L., Chen, S.Y., Jiang, K., Li, C., Lai, Y. and Fu, H., 2023. SketchFaceNeRF: Sketch-based facial generation and editing in neural radiance fields. ACM Transactions on Graphics, 42(4).

- [2] Sun, Y., Wu, Q., Zhou, H., Wang, K., Hu, T., Liao, C.C., Miyafuji, S., Liu, Z. and Koike, H., 2023, October. Make Your Brief Stroke Real and Stereoscopic: 3D-Aware Simplified Sketch to Portrait Generation. In Proceedings of the 25th International Conference on Multimodal Interaction (pp. 388-396).

**Suitability:**

2

---

### Official Review · Reviewer_HY4G · 2024-05-14

**Rating:** 4
**Confidence:** 4

**Summary:**

The authors propose a sketch-to-3D face method called S2TD-Face. S2TD-Face is able to reconstruct detailed geometries from facial sketches, and utilize CLIP to control the texture of the face. Qualitative and quantitative experiments illustrate the effectiveness of this method.

**Strengths:**

1. This paper is clearly written and easy to understand

2. The visualization quality of this paper is quite high.

**Limitations:**

1. I believe the primary goals of facial sketch reconstruction should be: 1) to recreate details challenging to capture in RGB images, such as wrinkles; 2) to maintain smoothness in the absence of these facial detail lines; and 3) to ensure the geometry of the reconstruction closely matches the contour lines of the sketches. However, the results presented in the paper fall short in these aspects.

2. Methodologically, the only technical contribution of this work appears to be the L_{sketch} loss function.

3. Is the texture control model truly necessary? Why not simply use SD and ControlNet?

4. Experimentally, the more appropriate baseline should be ControlNet + HRN, a method set that better recovers facial geometry in terms of contour lines compared to the proposed methods.

5. The authors utilized the approach outlined in Reference[59] to give additional supervision to the method, which likely led to their superior face alignment performance on the RELAY Benchmark.

**Suitability:**

3

---

### Official Review · Reviewer_xMmY · 2024-05-23

**Rating:** 6
**Confidence:** 2

**Summary:**

The paper presents a method named S2TD-Face for reconstructing the detailed 3D face from a single sketch image. Unlike existing approaches that rely on multiple transformation steps (e.g., two stages manner: i. mapping sketch image to 2D picture; ii. reconstruction 3D face geometry from 2D picture.), S2TD-Face directly predicts the coefficients of 3DMMs to reconstruct coarse geometry and then uses the coarse geometry and sketches in UV space to generate displacement maps for detailed face geometry. Besides, the method introduces a novel sketch-to-geometry loss function to supervise both coarse and detailed geometry reconstruction.

Extensive experiments show that S2TD-Face outperforms existing methods.  The ablation study as shown in Tab.1 also reveals the effectiveness of the proposed sketch loss.

**Strengths:**

1. Effective geometry reconstruction: S2TD-Face accurately captures the local details (e.g., wrinkles in the last row in Fig.1) of the input sketch. It achieves this by predicting 3DMM coefficients (mentioned as coarse geometry reconstruction) and optimizing a UV space displacement map.
2. Texture control module: S2TD-Face offers a texture control module that allows users to control the texture of the reconstructed 3D faces based on text prompts. This enables the generation of 3D faces with various styles ranging from cartoons to realistic appearances.
3. Superior performance: Extensive experiments show that S2TD-Face achieves excellent performance and outperforms existing methods on the Sketch-REALY benchmark. The qualitative results (e.g.Fig.6) show that the proposed method can capture the high-frequency details in the input sketch image.

**Limitations:**

1. Texture generation limitations: The textures generated by S2TD-Face are obtained from existing facial images (i.e. FFHQ). First, they find the closest 2D image picture in a predefined library (such as FFHQ) via prompts and image embedding matching in the CLIP embedding space.  Then, they use a texture reconstruction method [59] to obtain the texture map in the PCA space of the parametric 3D face model ( 3DMM). To alleviate the bias between the 2D image and the texture PCA space, they fuse the texture map in PCA space and the image corresponding UV space texture map.  This process may result in noticeable visual differences at the fusion boundaries. Besides, the diversity of the pre-defined image library limits the diversity of the generated texture map.

2. Realistic appearance limitation: The geometry reconstructed by S2TD-Face is derived from 3DMMs, which means that the overall appearance still resembles real human faces. This may limit the ability to handle animation styles that require non-human facial features.

**Suitability:**

3

---

### Official Review · Reviewer_vpmH · 2024-05-26

**Rating:** 3
**Confidence:** 3

**Summary:**

This paper proposes a method for reconstructing textured 3D faces from sketches and text prompt. The geometry reconstruction framework predicts the coefficients of 3DMMs from input sketches to reconstruct coarse geometry and then utilizes coarse geometry and sketches in UV space to generate displacement maps for detailed geometry. Then, a sketch-to-geometry loss is proposed to keep geometry consistent with the sketch. Finally, a text-image module is used to control the texture of reconstructed 3D faces based on text prompt. Extensive experiments show that it achieves excellent performance and outperforms the existing methods.

**Strengths:**

(1) This paper proposes a novel sketch-to-geometry loss to ensure both coarse and detailed geometry fidelity to the sketch input, and to maintain robustness across different sketch types.
(2) A novel texture control module for controlling the texture of the reconstructions, resulting in textured 3D faces with various styles.

**Limitations:**

(1) The five styles of sketches in figure 2 all appear to be synthetic, and I'm concerned about how well such training strategies will work for simple hand-drawn or stick-drawing style sketches. Additionally, reconstructions based on 3DMM may result in generated faces that lean towards realistic human faces, potentially lacking in representation of abstract styles.
(2) The qualitative comparison experiment selected in Figure 7 is limited, and such comparisons are somewhat inappropriate. Perhaps it would be better to select some more recent works as comparison baselines.
(3) Using CLIP text-image matching for semantic-level alignment might result in some details from the input sketches not being reflected in the textures, causing the synthesized final face to lose some information from the input sketch (e.g. the wrinkles on the forehead).

**Suitability:**

2

---

### Meta-Review · Area_Chair_oEBA · 2024-07-01

**Recommendation:** Accept (Poster)
**Confidence:** 4

**Metareview:**

This paper presents a method named S2TD-Face for reconstructing the detailed 3D face from a single sketch image.
The Reviewers did not find an agreement on the paper. Two of them tended toward accepting the paper, founding the proposed method with novel contributions. In my views, according to the reviews, the reasons for accepting the paper are prevailing on those for rejecting it. So my suggestion is to have the paper accepted as a poster.